# Quality of Life (QoL) in Patients with Chronic Inflammatory Bowel Diseases: How Much Better with Biological Drugs?

**DOI:** 10.3390/jpm13060947

**Published:** 2023-06-02

**Authors:** Federica Bellone, Carmela Morace, Giulia Impalà, Anna Viola, Alberto Lo Gullo, Maria Cinquegrani, Walter Fries, Alberto Sardella, Mariangela Scolaro, Giorgio Basile, Giovanni Squadrito, Giuseppe Mandraffino

**Affiliations:** 1Internal Medicine Unit, Department of Clinical and Experimental Medicine, University of Messina, 98122 Messina, Italy; 2Unit of Anaesthesia and Intensive Care, Department of Clinical-Surgical, Diagnostic and Pediatric Sciences, University of Pavia, 27100 Pavia, Italy; giulia.impala01@universitadipavia.it; 3IBD Unit, Department of Clinical and Experimental Medicine, University of Messina, 98122 Messina, Italy; anna.viola@unime.it (A.V.);; 4Unit of Rheumatology, Department of Medicine, ARNAS Garibaldi Hospital, 95124 Catania, Italy; albertologullo@virgilio.it; 5Department of Clinical and Experimental Medicine, University of Messina, 98122 Messina, Italy; 6UOC Cardiologia/Utic Ospedale G. Fogliani Milazzo Asp 5, 98123 Messina, Italy; 7Department of Biomedical and Dental Sciences and Morphofunctional Imaging, University of Messina, 98122 Messina, Italy; basileg@unime.it

**Keywords:** quality of life, inflammatory bowel disease, Crohn’s disease, ulcerative colitis, biologic drugs, infliximab, vedolizumab, deep remission, multidimensional health status

## Abstract

Background: Inflammatory bowel diseases (IBDs), including ulcerative colitis (UC) and Crohn’s disease (CD), are chronic and disabling diseases that affect patient health-related quality of life (HRQoL). IBD patients are frequently exposed to high levels of stress and psychological distress. Biological drugs have been proven to reduce inflammation, hospitalization, and most of the complications that characterize IBDs; their potential contribution to patients’ HRQoL remains to be explored. Aim: To evaluate and compare any change in the HRQoL and markers of inflammation in IBD patients undergoing biological drugs (infliximab or vedolizumab). Material and Methods: A prospective observational study was conducted on a cohort of IBD patients, aged >18 years, who were prescribed with infliximab or vedolizumab. Demographic and disease-related data at baseline were collected. Standard hematological and clinical biochemistry parameters, including C-reactive protein (CRP), white blood cells count (WBC), erythrocytes sedimentation rate (ESR), and α1 and α2 globulins were measured after a 12-h fast at baseline (T0), after 6 weeks (T1), and at 14 weeks (T2) of biological treatment. Steroid use, disease activity as measured by the Harvey–Bradshaw index (HBI) and partial Mayo score (pMS) for the CD and UC, respectively, were also recorded at each timepoint. The Short Form 36 Health Survey (SF-36), Functional Assessment of Chronic Illness Therapy (FACIT-F), and Work Productivity and Activity Impairment–General Health Questionnaire (WPAI:GH) were administered to each patient at baseline, T1, and T2 to address the study aims. Results: Fifty eligible consecutive patients (52% with CD and 48% with UC) were included in the study. Twenty-two patients received infliximab and twenty-eight received vedolizumab. We noted a significant reduction in the CRP, WBC, α1, and α2 globulins from T0 to T2 (*p* = 0.046, *p* = 0.002, *p* = 0.008, and *p* = 0.002, respectively). Participants showed a significant decrease in steroid administration during the observation period. A significant reduction in the HBI of CD patients at all three timepoints and a similarly significant decrease in the pMS of UC patients from baseline to T1 were recorded. Statistically significant changes were observed in all questionnaires during follow-up as well as an overall improvement in the HRQoL. The interdependence analysis carried out between the biomarkers and the scores of the individual subscales showed a significant correlation between the variation (Δ) of the CRP, Hb, MCH, and MCV with physical and emotional dimensions of the SF-36 and FACIT-F tools; work productivity loss expressed by some of the WPAI:GH items negatively correlated with the ΔWBC and positively with the ΔMCV, ΔMCH, and Δ α1 globulins. A sub-analysis according to the type of treatment showed that patients receiving infliximab experienced a more pronounced improvement in their HRQoL (according to both SF-36 and FACIT-F) compared with patients receiving vedolizumab. Conclusions: Both infliximab and vedolizumab played an important role in contributing to the improvement of the HRQoL in IBD patients by also reducing inflammation and, consequently, steroid use in patients with an active disease. HRQoL, being one of the treatment goals, should also be assessed when taking charge of IBD patients to assess their clinical response and remission. The specific correlation between the biomarkers of inflammation and life’s spheres, as well as their possible role as clinical markers of HRQoL, should be further investigated.

## 1. Introduction

Inflammatory bowel diseases (IBDs) are mainly represented by Crohn’s disease (CD) and ulcerative colitis (UC), which are two idiopathic chronic diseases that are characterized by mucosal intestinal inflammation; the diseases affect both children and adults with spreading incidence and prevalence in recent decades [1]. Different factors contribute to complex IBD pathogenesis, involving genetic predisposition, mucosal immune response, gastrointestinal microbiota alterations, and environmental and lifestyle factors; a better understanding of IBD etiopathology has been promoted to improve and tailor the therapeutic management [2].

A wide range of new therapeutical options for IBD patients is currently available. The conventional pharmacological approach includes aminosalicylates, corticosteroids (CSs), and immunomodulators (azathioprine, 6-mercaptophurine, methotrexate); the introduction of anti-tumor necrosis factor (TNF) agents (infliximab, certolizumab pegol, adalimumab, and golimumab) has allowed for the achievement of a high rate of long-lasting remission and an impressive change of the disease course [3]. Nevertheless, evidence of non-response in a proportion of IBD patients has raised the need for new treatment strategies; anti-integrin molecules (natalizumab and vedolizumab) have been more recently approved for IBD treatment [4].

The common purpose of this wide range of pharmacological therapy for IBDs is to suppress abnormal inflammatory response, fixing the immune dysregulation [5]. Indeed, because biological therapies have been introduced, the goal of IBD treatment has changed from simple clinical remission—or avoidance of surgery—to deep remission [6]. However, patient-reported outcomes, quality of life (QoL), and related psychosocial measures have acquired increasing interest in the last years, and they have also become clinical outcomes in randomized clinical trials [7]. Nowadays, one of the main goals of IBD therapy is to improve the patients’ health-related quality of life (HRQoL) by reducing inflammation [8,9].

The traditional treatment approach for IBDs includes a step-up approach from conventional treatment—as mesalamine, corticosteroids, and immunomodulators—to biological drugs (anti-TNF, anti-integrine, and anti-interleukine) or the newest small molecules that target JAK (tofacitinib). Typically, 5-aminosalicylic acid (5-ASA) is used for the induction and maintenance of the remission of mild-moderate active UC [10], and corticosteroids are used to induce remission in severe disease or during a flare for both UC and CD. Patients who fail to be treated using conventional treatment or who have a severe disease should be treated using biological drugs, which could also be considered as an early treatment in patients with poor prognostic factors.

Disease activity and treatment response are assessed using clinical and endoscopic scores, biochemical biomarkers of inflammation, and more recently using therapeutic drug monitoring (the measurement of drug and/or anti-drug antibody levels to assess compliance, drug metabolism, and immunogenicity). In routine practice, several biomarkers are used with the aim of detecting inflammation during the patients’ follow-ups. The best-known and most frequently used markers of inflammation are C-reactive protein (CRP) and fecal calprotectin (FC). CRP is an acute phase protein that increase in response to inflammation, and it is the most frequently used serum biomarker in clinical practice because of its low cost and non-invasiveness. Other serum markers of the acute phase response are the erythrocyte sedimentation rate (ESR), platelets count (PLT), red blood cell distribution (RDW), and white blood cells (WBCs) count [11,12]. All of these markers are increased during the active phase of IBDs. The normalization of inflammation markers are the basis for monitoring the treatment response together with clinical scores and endoscopic activity.

Biologic therapies were confirmed to be effective at achieving clinical response and remission in IBDs and are currently being questioned on if they improve the HRQoL [13]. The benefit of biological therapy on quality of life could be related to the treatment response and reduction in inflammation; however, there are no data on the specific correlation between improvements in the HRQoL and the normalization or reduction of biomarkers during treatment.

In the present study, we aimed to investigate the objective correlation between markers of inflammation and the HRQoL in patients with active UC or CD who are undergoing intravenous biological therapies.

## 2. Materials and Methods

### 2.1. Participants

In this monocentric, prospective, observational study, we collected the data of patients with a confirmed clinical, endoscopic, and/or histological diagnosis of UC and CD, who were ≥18 years of age and who started taking infliximab and vedolizumab from October 2018 to May 2019. The patients included had to be able to sign an informed consent and had to understand and complete all questionnaires offered.

Infliximab and vedolizumab were prescribed in patients with active moderate-to-severe CD and UC. IV infusion for infliximab was weight-based (5 mg/kg) at week 0 followed by an infusion at week 2 and week 6 to complete induction; a maintenance was performed every 8 weeks according to the patient’s response. IV infusion for vedolizumab was 300 mg at week 0, at week 2, at week 6, and every 8 weeks as maintenance according to the patient’s response. The dosage and timing for drug prescription and administration were planned according to the recommended treatment schedule(s).

The treatment response was evaluated as in clinical practice during the induction and every 2 months during maintenance using biochemical tests and a clinical assessment.

### 2.2. Data Collection and Outcome Measures

The following data were collected at baseline for each patient: sex, age at enrolment and at diagnosis, type of disease, previous surgery, concomitant steroids, and background therapy. Patients were classified in accordance with the Montreal classification with respect to disease extent.

Moreover, the disease activity at baseline and during follow-up was assessed for UC and CD using the partial Mayo score (pMS) and Harvey–Bradshaw index (HBI), respectively. The endoscopic activity at baseline was assessed using the Simple Endoscopic Score for Crohn’s disease (SES-CD) and Rutgeerts score for patients with CD. The endoscopic Mayo score was used for patients with ulcerative colitis. Data on steroid use at different timepoints were also collected. The primary outcome was the correlation between the markers of inflammation and the HRQoL, which was measured using the Short Form-36 Health Survey (SF-36), Functional Assessment of Chronic Illness Therapy (FACIT-F), and the Work Productivity and Activity Impairment: General Health Questionnaire (WPAI:GH) tools.

All questionnaires were administered to each patient at the following timepoints: baseline (T0), 6 weeks after (T1), and 14 weeks after (T2) the start of the biological treatment.

The hematological and clinical biochemistry parameters (ESR, CRP, alfa1-alfa2 globulin, and blood count) were measured after a 12-h fast during the same timepoints.

### 2.3. Measurements

The perceived HRQoL was assessed using the Italian version of SF-36 [14,15] (license-free from Rand Corporation). SF-36 is a self-report questionnaire that explores eight health-related domains, namely perceived mental health, emotional role, social functioning, vitality, general health, body pain, physical role, and physical functioning. The SF-36 scores range from 0 to 100; higher scores are an expression of a greater perceived HRQoL.

According to the free-of-charge license for the Italian version of the questionnaire provided by FACIT.org, the FACIT questionnaire was also administered in order to assess the multidimensional health status of people with chronic illnesses [16]. The FACIT-F is a self-report, forty-item questionnaire; the items are rated on a 5 point Likert scale, ranging from 0 (“not at all”) to 4 (“very much”), based on the perceived impact of certain symptoms on the individual’s daily life. The FACIT-F is composed of 5 sub-scales, measuring “Physical well-being” (score ranging from 0 to 28), “Social/family well-being” (score ranging from 0 to 28), “Emotional well-being” (score ranging from 0 to 24), “Functional well-being” (score ranging from 0 to 28), and “Additional Concerns” (score ranging from 0 to 52). Indeed, we assessed fatigue and its impact on daily activities and functioning through the use of the fatigue sub-score (FACIT-FS), which is based on the score obtained in the last sub-scale of the FACIT [17]; lower scores on the FACIT-FS are expressions of greater perceived fatigue. According to the study’s aim, we used both the FACIT score from the four scale (FACIT-G) and the global FACIT score (FACIT-GH), which is obtained by summing the FACIT-G with the FACIT-FS.

Additionally, the WPAI:GH tool was administered [18] (as provided by Margaret Reilly Associates Inc.; Italian version). The WPAI:GH is a 6 item questionnaire thatmeasures the effect of health problems (e.g., physical, emotional) on the individual’s ability to work and perform regular activities. More precisely, the questionnaire first asks about the number of days and hours missed from work, the days and hours effectively worked, the days during which performing work was difficult, and the extent to which the individual was limited at work during the past 7 days. In addition, the WPAI:GH evaluated the impairment in regular activities beyond work (e.g., shopping, studying, childcare); the items are rated on a 10 point visual analog scale, ranging from 0 (“Health problems had no effect on my daily activities”) to 10 (“Health problems completely prevented me from doing my daily activities”).

## 3. Statistical Analysis

The variables distribution was verified using the Kolmogorov–Smirnov test; according to the non-normal distribution of some variables and the relatively small sample size, a conventional non-parametric statistical approach was chosen. Consistently, the numerical data were expressed as the median and IQR, and categorical variables were expressed as the number and percentage. Repeated measures were tested using Friedman’s test, while dichotomous variables were tested using Cochran’s test. Any change over time was verified using the Wilcoxon signed-rank sum test (including the variation of biochemical tests and subscale of the three questionnaires), and the between-groups difference was tested using the Mann–Whitney test.

Spearman’s test was carried out to assess the interrelationship between the variation (Δ) of biochemical tests with the Δ of the single subscales (T2-T0). *p*-value < 0.05 was considered to be statistically significant. The statistical analysis was performed by using the SPSS statistical package, version 26.0 (Chicago, IL, USA).

## 4. Ethical Statement

Each procedure completed in this study was in accordance with the ethical standards of the local institutional research committee, and with the 1964 Helsinki Declaration and its later amendments. Written informed consent was collected for all participants. The study was approved by the local ethics committee under protocol no. 66-19.

## 5. Results

Fifty IBD patients were consecutively enrolled. The clinical and biochemical characteristics at baseline (T0), T1, and T2 of the recruited population are shown in Table 1. Twenty-six patients (52%) were diagnosed with CD, whereas 24 patients (48%) were diagnosed with UC. Overall, 44% (*n* = 22) of patients received IV infusions of infliximab and 56% (*n* = 28) received IV infusions of vedolizumab. No other biological therapies had been previously started by the enrolled patients. A large amount of the participants (76%, *n* = 38) had never underwent IBD surgery and did not in the 14 weeks after they started biological treatment.

A recent endoscopy (latest six months) at baseline was available in 42/50 patients (84%), and active disease (moderate-to-severe) was confirmed in all of them. In detail, patients with CD had a median SES-CD of 18 (7–30) and a median Rutgeerts score of 4 (2–4), and patients with UC had a median endoscopic Mayo score of 2 (2–3).

A recent endoscopy was not available for seven patients. Among them, two patients had instrumental signs (MRI or bowel ultrasound) of increased disease activity.

Additionally, none of the recruited patients required oral iron supplementation and/or blood transfusion during the observation.

After the administration of a biological treatment, a significant reduction in inflammatory indices was found; particularly, the CRP, WBC, α1, and α2 globulins serum levels were significantly reduced from T0 to T2 (*p* = 0.046, *p* = 0.002, *p* = 0.008, and *p* < 0.001, respectively; (Table 2).

A significant decrease in the need for steroid therapy was observed at each timepoint in the whole study population, as depicted in Figure 1.

A clinical response was also recorded in CD participants, which was expressed in terms of a reduction in the HBI (T1 vs. T0 *p* = 0.001; T2 vs. T1 *p* = 0.109; T2 vs. T0 *p* = 0.017) (Figure 2).

The same reduction in disease activity as assessed by the pMS was observed in UC patients, as represented in Figure 3 (T1 vs. T0 *p* = 0.001; T2 vs. T1 *p* = 0.31; T2 vs. T0 *p* = 0.44).

A statistically significant improvement in the perceived HRQoL as assessed by the SF-36, FACIT:GH, and WPAI:GH questionnaires was reported during the follow-up period. (Table 3).

In detail, in all SF-36 items, a statistically significant positive change was detected at the end of the observation period compared with the baseline. A statistically significant improvement only emerged in three of the eight items when comparing all timepoints (Appendix A).

The same result was detected for the FACIT:GH questionnaire except for the social/family well-being item (Appendix A).

WPAI:GH scores significantly improved: Out of the 50 selected patients, 23 were working at baseline; 9 patients were retired from work, and 18 were not workers (Appendix A). Four patients started working between baseline and T1, while three patients stopped working within the same time frame.

With respect to the hours of working time missed due to health issues, we noted a significant reduction over time (*p* = 0.018 from T0 to T2); additionally, the degree of work productivity affected by the disease has consistently significantly reduced (*p* < 0.001) as well as the degree of impairment of regular activities (*p* < 0.001).

We also carried out an interdependence analysis in order to assess the interrelationships between the change in serum biomarkers and the scores of the individual subscales of the SF-36, FACIT-F, and WPAI:HG questionnaires. In detail, significant associations emerged between SF-36 items (physical functioning, physical health, and emotional role) variations and CRP serum levels changes from T0 to T2. Interestingly, the change over time of the “physical health” item of the SF-36 questionnaire showed a significant correlation with the Δ of more inflammatory indices (WBC, neutrophils, lymphocytes, CRP). Conversely, the modification of Hb, MCH, and MCV were inversely associated with the changes in emotional role, body pain; the last two variables were inversely associated with general health. With respect to the FACIT-F questionnaire, a significant association between the variation in CRP values and physical well-being and FACIT-G items were found, along with negative correlations between MCV changes and physical well-being, FACIT-G items, and FACIT-GH variations. The WPAI:GH questionnaire showed non-significant correlations between the variation in CRP values and its items; conversely, a negative statistically significant association emerged between WBC and changes in work time missed (other), effective worked hours, and impairment at work. A significant positive association between ΔMCV and ΔMCH and the variation in the work time missed (health) item was detected; interestingly, changes in alpha1 globulins serum levels positively correlated with the Δ of the work time missed (other) item of the WPAI:GH test (Table 3).

As shown in Table 4, besides the CRP, Hb, MCV, and MCH, no additional patterns of dependence with other inflammatory indices were identified, which was likely due to the small sample size. In addition, the Δ of all SF-36 items showed patterns of dependence with each other, as well as for the FACIT items. In turn, the two HRQoL questionnaires also showed significant correlation matrices with each other (Appendix A). The interrelationships among serum parameters were also verified by the Spearman’s test, and the results were consistent with what was expected (see details in Table 4); the whole correlation panel is reported in Appendix A.

We also tested the interrelationships between the HRQoL scores and HBI (CD patients) or pMS (UC patients) values. We found a significant correlation between HBI scores and: the “physical functioning” and “physical health” items of the SF-36 questionnaire (rho −0.414, *p* < 0.05; rho −0.334, *p* < 0.05, respectively); and the “work time missed (health)”, “impairment at work” and “regular activity impairment” items of the WPAI:HG test (rho 0.644, *p* < 0.001; rho 0.408, *p* < 0.05; rho 0.457, *p* < 0.01, respectively). With respect to the UC patients, we found a significant correlation between pMS scores and: the “additional concerns (FACIT-FS)” item of the FACIT questionnaire (rho −0.347, *p* < 0.05); the “work time missed (health)”, “impairment at work”, and “regular activity impairment” items of the WPAI:HG questionnaire (rho 0.359, *p* < 0.01; rho 0.574, *p* < 0.005); rho 0.344, *p* < 0.05, respectively.

A sub-analysis according to the type of therapy (infliximab vs. vedolizumab) showed that patients receiving infliximab experienced a more pronounced improvement in the perceived HRQoL (according to SF-36 and FACIT-F) compared with patients receiving vedolizumab: specifically, the vitality, social functioning, and body pain SF-36 domains and physical well-being and additional concerns (FACIT-FS) items of the FACIT-F questionnaire showed a greater significant change (Δ) in patients that were treated with infliximab compared with those treated with vedolizumab (Table 5 and Table 6; Figure 4). Indeed, when we tested the HBI and pMS change over time according to the type of therapy, we found no significant difference (*p* = 0.618 and *p* = 0.343, respectively).

## 6. Discussion

Inflammatory bowel diseases (IBDs) are chronic, progressive, and disabling conditions that are characterized by chronic, uncontrolled, and relapsing inflammation of the gastrointestinal tract [19]. IBD symptoms negatively affect patients’ social and daily functioning, thus unavoidably impacting their psychological and social well-being [20]. Furthermore, IBD patients are more inclined to have the inability to recognize and express emotions, showing greater alexithymic levels [21]; therefore, the psychopathological comorbidity may be underestimated. Our study sample was characterized by a worse perceived HRQoL in both the physical and mental domains, thus highlighting the underlying psychological suffering that characterizes IBDs. Available treatments for IBDs are mainly aimed at reducing disease-related inflammation [7]; the drugs that are commonly administered include immunosuppressants and steroids. These drugs cause some side effects, such as an increased risk of infection and impaired body image, which in turn increases the psychological burden and reduces patients’ medication adherence, negatively affecting overall HRQoL [22]. It is well documented that treatment with biological agents improves clinical outcomes and leads to patient satisfaction and a better HRQoL [13]. However, studies on the biological treatment and HRQoL are widely heterogeneous due to the patients included and questionnaires employed. The present study aimed to deepen the insight on the potential changes in the perceived HRQoL and the relationships between the treatment response (both clinical and biochemical), steroid sparing, and a better HRQoL in IBD patients; this aspect represents the main strength of our study.

The patients, as evaluated before and after the induction of infliximab and vedolizumab, had an improvement in their HRQoL according to all three of the questionnaires that were conducted. With a similar intent, Burisch et al. investigated HRQoL using the Short Inflammatory Bowel Disease Questionnaire (SIBDQ) and Short Form 12 (SF-12) questionnaires but on a heterogeneous population of 1079 UC and CD patients, restricting their research to the first year of disease and not excluding those patients eventually undergone surgery [23]; our study investigated - in line with most placebo-controlled trials-IBD patients with a long disease duration. Another strength of our study is the comparison of two biologics rather than biologics versus placebo. In the above-mentioned paper, biological treatment improved the HRQoL in CD patients, while UC patients in need of surgery or biological therapy experienced lower perceptions of HRQoL than the rest [23]; this observation was examined without focusing on the effect of individual biologics. A recent systematic review by Aladraj et al. only focused on CD patients reported from 16 RCTs about the superior efficacy of biological and small-molecule drugs for improving the HRQoL outcomes. Nevertheless, in almost all of the considered studies (15/16), the investigated pharmacological interventions were compared to a placebo [24]; as a result, due to the paucity of the comparative analysis of biologics and small-molecule drugs with other pharmacological agents in the published literature, our study gains originality.

The improvement in the HRQoL seems to be closely related to the treatment response and consequently to the clinical response and remission. We observed a significant improvement in disease activity scales after biologic therapy was started for both CD (HBI) and UC (pMS) patients. This improvement was consistent with the perceived quality of life, as evaluated by the SF-36, FACIT, and WPAI:HG questionnaires. As already reported elsewhere, both the HBI and pMS scores improved from T0 to T2, which was through the T1 intermediate timepoint and after biologic therapy was started. However, the activity scores (both the endoscopy- and clinical-based ones) are different in CD and UC patients, and they are not interchangeable.

Clinical scores are commonly used in clinical practice for the clinical response assessment. There is a strong agreement between different clinical scores, patient reported outcomes (PROs), and endoscopic activity, especially for UC [25]. HRQoL scores are shown to be useful in both bowel inflammatory diseases; moreover, Tribbick et al. reported a significative association of increased disease activity, assessed through the Manitoba index (MI), with more severe anxiety and depression and a reduced quality of life in female patients, which is closely related to illness perception [26].

We also observed a certain correlation between the considered HRQoL questionnaires and the clinical scores; this correlation could support the use of HRQoL scores to verify the response to therapy along with clinical scores. Indeed, we can confirm only a few correlations between the items that assessed quality of life and the HBI/pMS scores. This could be explained by the different aims of the different score systems and the more objective measurement of disease activity from the HBI and pMS scores (including physician rating of disease activity). Questionnaires assessing HRQoL are not created to evaluate disease activity but are rather indirect indicators of clinical well-being and a direct estimate of the patient’s quality of life which, in recent years, has become one of the main therapeutic goals.

With this in mind, HRQoL questionnaires could be confirmed as simple tools for use in clinical practice to evaluate the perceived quality of life along with the clinical–biologic response.

These results are in line with the current evidence on HRQoL in patients treated with biologics. Zhang et al. measured illness perceptions, coping strategies, anxiety, depression, and quality of life (measured using the Inflammatory Bowel Disease Questionnaire (IBDQ)) during treatment with infliximab in a cohort of CD patients. There was a significant improvement in the illness perception, maladaptive coping, anxiety, depression, and HRQoL in patients with clinical remission (according to HBI score) at week 14 and 30, while no significant changes were observed in non-responder patients. The impact of steroids was not evaluated in the above-mentioned paper.

Steroids are known to cause many adverse effects, which are particularly related to the CS dosage and long-term exposure; they mainly consist of diabetes mellitus, metabolic syndrome, weight gain, hypertension, gastrointestinal bleeds/ulcers, cataracts, glaucoma, increased risk of infections, compromised wound healing, muscle weakness, and psychological disorders [27]. The latter are represented by mood disorders such as irritability, insomnia, anxiety, depression, and psychosis; the underlying mechanism by which CSs induce psychological symptoms is multifactorial and is probably due to either direct or indirect effects on the brain system [28]. The consequences that some physical CS side effects, such as fractures or weight gain, can have on the psychological sphere are also not excluded [29]. This inevitably worsens the HRQoL of patients who are undergoing steroid therapy. To date, little research has investigated the impact of taking CSs on the HRQoL apart from the clinical disorder for which they are taken. Sullivan et al. assessed the HRQoL of patients who were using systemic CSs, demonstrating that more than four prescriptions of systemic CSs per year were associated with a significantly lower HRQoL, as measured by health questionnaire (EQ-5D) and SF-6D indices [30]. On the basis of our findings, we can therefore hypothesize that the HRQoL improved in our study cohort due to the improvement in disease symptoms and bleeding following the start of the biologic drugs, as highlighted in our research by the improvement of clinical scores; reducing the steroid needs could in turn further improve the perceived quality of life [27,28]. Quality of life has been also evaluated in several RCTs on biological drugs and small molecules. In our study, both of the biologic drugs used in this population confirmed their significant impact on disease activity and perceived quality of life; however, we recorded a less extended effectiveness of vedolizumab compared with infliximab.

Although the small sample size does not allow us to generalize the results, we found a difference in the impact of the treatment type on some HRQoL items, with a greater improvement related to infliximab administration compared with vedolizumab, at least within the first 14 weeks. This significant improvement observed in a closer timeframe seems to be remarkable, as the early restoration of HRQoL has been shown to predict long-term remission in infliximab-treated patients [31].

Data on vedolizumab’s impact on QoL are available from clinical trials for both vedolizumab’s intravenous formulation and the more recent subcutaneous formulation [32,33]. Indirect comparisons between different biological drugs in terms of their impact on QoL are also available from reviews and network meta-analyses. Paschos et al. evaluated 14 RCTs and assessed the HRQoL in UC patients using the Inflammatory Bowel Disease Questionnaire (IBDQ), SF-36, or European Quality of Life-5 Dimensions questionnaire (EQ-5D). Among all of the biologics evaluated, (infliximab, vedolizumab, golimumab, adalimumab, and tofacitinib) infliximab (MD 18.58; 95% CI 13.19–23.97) and vedolizumab (MD 18.00; 95% CI 11.08–24.92) achieved the best improvement in mean IBDQ score compared with placebo after induction. Moreover, a greater improvement in the mean physical and mental component score of SF-36 has been observed in patients treated with infliximab, vedolizumab, and tofacitinib compared with placebo [34]. We also reported a significant improvement in all dimensions of the SF-36 tool during follow-up for the entire cohort. However, when a comparison between IFX and VEDO was performed, we showed a greater significant change (Δ) in patients treated with infliximab compared with those treated with vedolizumab. A possible explanation of this difference could be the rapidity of action of IV anti-TNF alpha compared with vedolizumab; however, the sample is too small to confirm this data.

Nowadays, the interest in fatigue has considerably increased as its prevalence reaches up to 86% of patients with active diseases [35,36]. The etiology of fatigue is poorly understood, but factors as inflammation, physical deconditioning, and nutritional and psychosocial factors seem to be involved. Data on the impact of biological therapies on fatigue are scarce and often contrasting [35,37]. We showed that a significant modification in the FACIT items were related to a decrease in inflammation and that the benefit was greater for infliximab than vedolizumab.

Inflammation plays a key role in IBD development and progression, which is the case in other diseases as well. A few reports have shown the possible link and correlation between systemic inflammation (evaluated through serum biomarkers) and specific diseases (eg. cancer, post-traumatic stress syndrome, SARS-CoV-2, etc.) [38,39,40]. Some of these papers have been explicitly focused on the relationship between CRP and HRQoL. Other studies have also focused on the link between HRQoL and interleukins. The inflammatory process could potentially be a contributing factor to poor HRQoL in different subset patients. In this setting, we hypothesized that HRQoL impairment in IBD patients may be related to their inflammation status (evaluating specific inflammation biomarkers (endocan, CRP, IL etc.)); this could be considered as a future study object.

Regarding our study population, a clarification should be added: more than half of the patients included were non-workers at baseline, which was not for disease-related concerns (16% were already retired from work and 36% were looking for first employment); this limitation could affect the large-scale reproducibility of the impact estimated on work productivity impairment. Indeed, our data are in line with the employment data from the Italian National Institute of Statistics (ISTAT) of the Sicilian population in relation to the year of the participants’ recruitment. In fact, the number of Sicilian households with no one employed was markedly higher than the national average in 2018 (specifically, 32.5% versus 18.4% percent in the whole Italian area, no. 479/1474).

The current research has some inherent limitations that should be acknowledged, such as the small sample size and some missing data in the patients’ medical records. Moreover, answers provided to the WPAI:GH questionnaire from participants to assess their health-related productivity loss lack uniformity, resulting in a consequent inability to perform correlation matrices; thus, our findings may not be generalizable. Another weakness of the study is the use of self-report questionnaires. We also recognize that all participants were referred to a single university outpatients’ clinic and that it could represent a selection bias; nevertheless, this could also be considered a strength due to the homogeneous treatment approach. Our longitudinal study investigated the change of perceived HRQoL over time and the main biomarkers of inflammation in a setting of IBD patients undergoing biological treatment for the first time; the results show a significant improvement in patients’ well-being and an earlier therapeutic effect of infliximab compared with vedolizumab. In addition to this undisputed strength, the longitudinal design of the current study confers more robustness to the research that we have carried out.

## 7. Conclusions

The present study provides—for the first time—evidence of the important role of both infliximab and vedolizumab in improving HRQoL by decreasing inflammation serum biomarkers and clinical scores in patients with an active IBD disease.

Due to the importance of psychological distress in altering IBD illness behavior and its negative effect on inflammation parameters and patient HRQoL, the integration of psychological strategies into conventional medical therapy seems to be recommended. Thus, our data could be practical for planning psychological strategies specifically designed for the management of psychological health concerns and for integrating them into conventional IBD medical treatment.

The specific correlation between inflammatory indices and both the physical and mental domains of HRQoL, as well as their potential role as clinical markers of IBD patients’ well-being, require further studies.

## Figures and Tables

**Figure 1 jpm-13-00947-f001:**
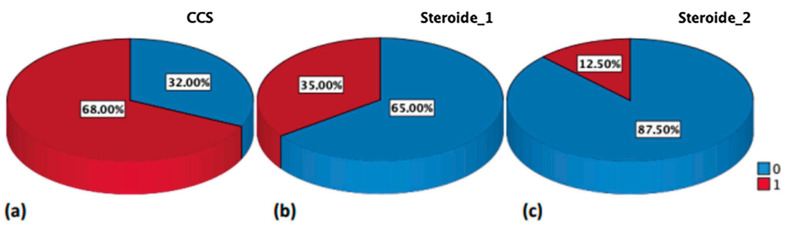
Corticosteroids need (% of patients) at baseline [panel (**a**)], T1 [panel (**b**)], and T2 [panel (**c**)], respectively; 0 (blue) = no steroids use; 1 (red) = steroids use.

**Figure 2 jpm-13-00947-f002:**
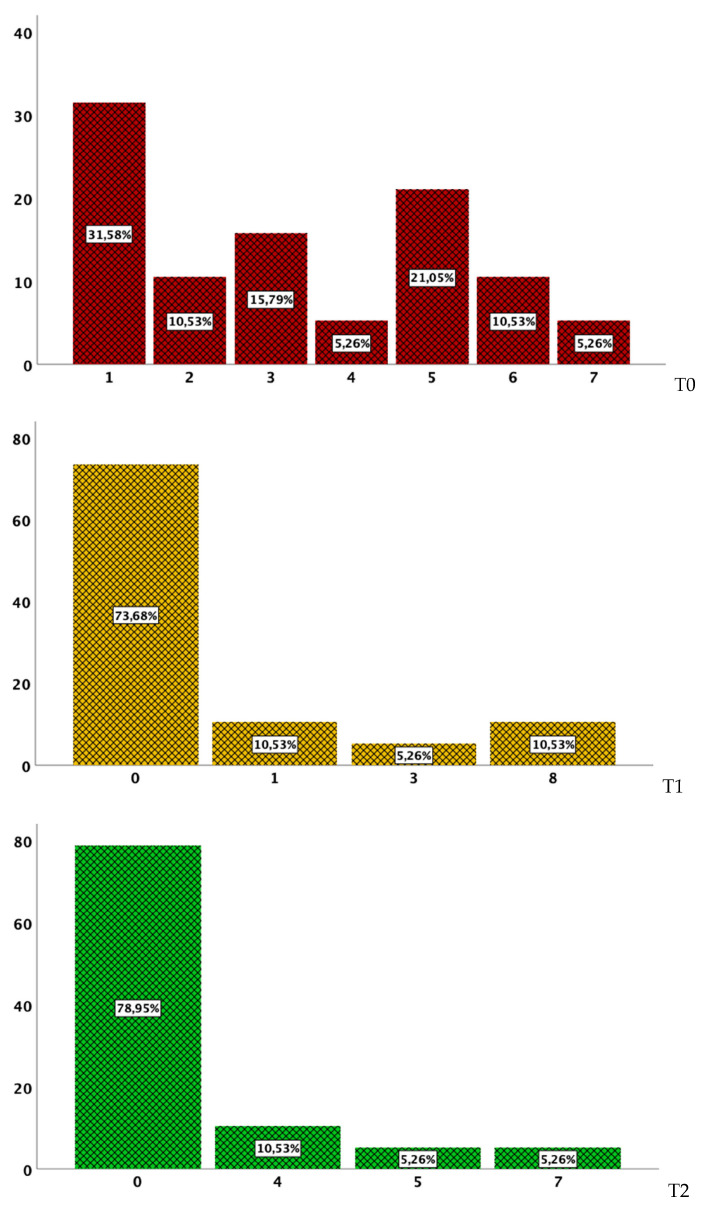
Frequencies (%) for the HBI score from baseline (red bars) to T1 (yellow bars) and T2 (green bars).

**Figure 3 jpm-13-00947-f003:**
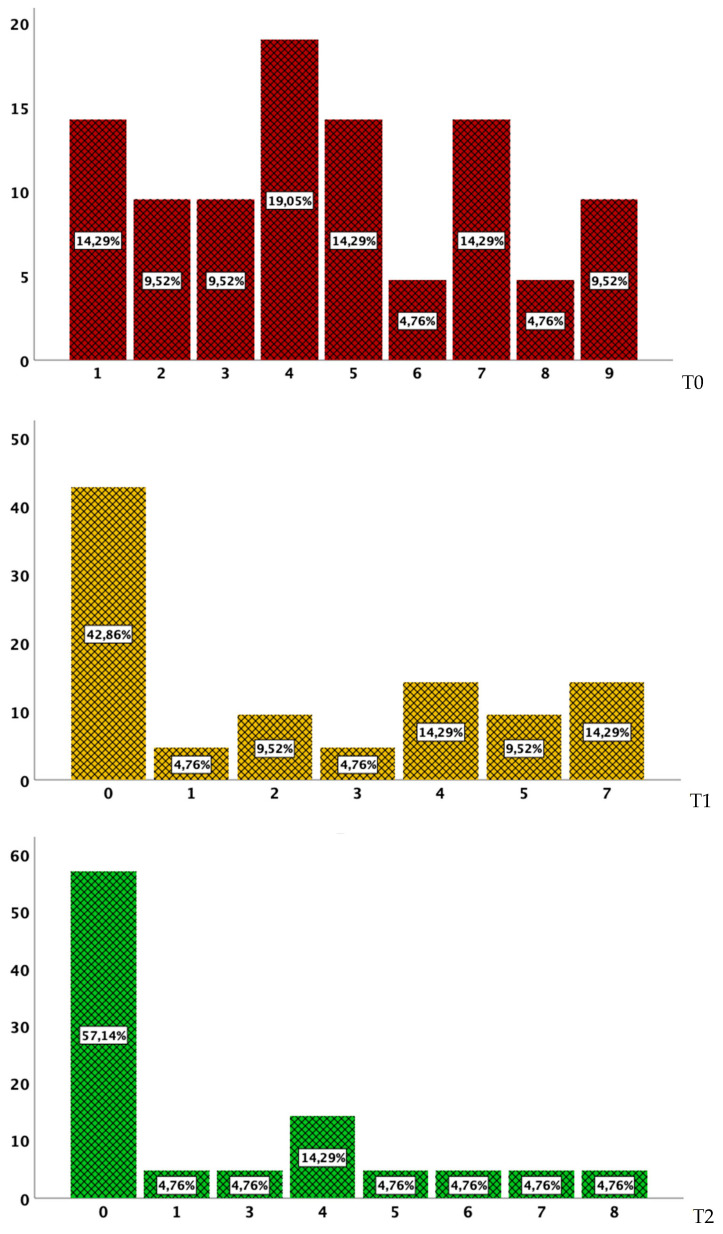
Frequencies (%) for the pMS score from baseline (red bars) to T1 (yellow bars) and T2 (green bars).

**Figure 4 jpm-13-00947-f004:**
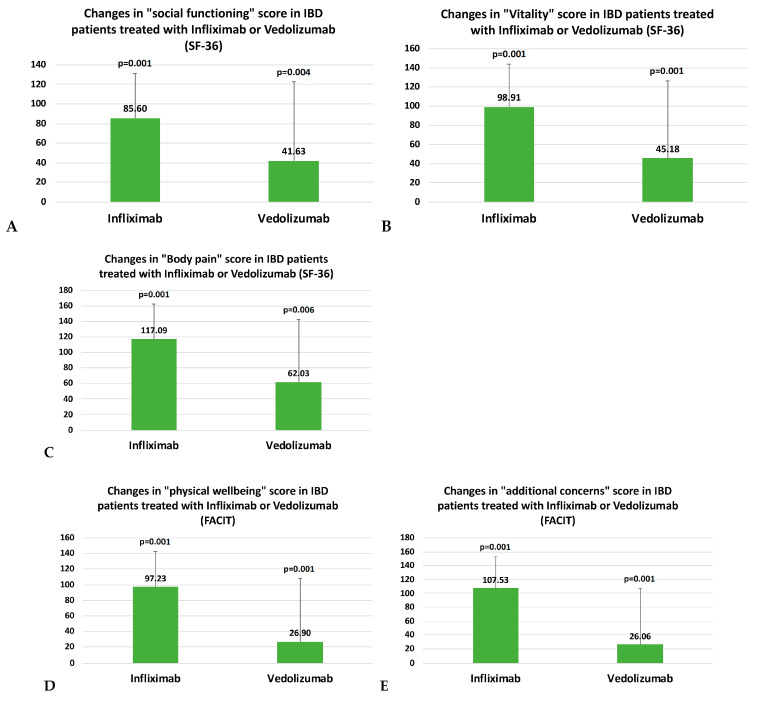
Histograms describing the change in SF-36 and FACIT-F item values from baseline (T0) to week 14 (T2) according to the type of treatment (infliximab, I; vedolizumab, V). Bars = delta value (%). Panel (**A**): social functioning SF-36 item: (I) *p* = 0.001 vs. baseline; (V) *p* = 0.004 vs. baseline. Panel (**B**): vitality SF-36 item: (I) *p* < 0.001 vs. baseline; (V) *p* = 0.001 vs. baseline. Panel (**C**): body pain SF-36 item: (I) *p* < 0.001 vs. baseline; (V) *p* = 0.006 vs. baseline. Panel (**D**): physical well-being FACIT-F item: (I) *p* < 0.001 vs. baseline; (V) *p* < 0.001 vs. baseline. Panel (**E**): additional concerns (FACIT-FS) FACIT-F item: (I) *p* < 0.001 vs. baseline; (V) *p* = 0.001 vs. baseline.

**Table 1 jpm-13-00947-t001:** Demographic, laboratory, and clinical characteristics of the sample at T0, T1, and T2.

	T0	T1	T2	*p*
Number	50	50	50	
Sex (m/f)	26/24	26/24	26/24	
Disease (CD/UC)	26/24	26/24	26/24	
Age at diagnosis	28 (12–44)			
Age at time of start of biological treatment	48 (33–63)			
No surgery/surgery	38/12	38/12	38/12	
Vedolizumab/Infliximab	22/28	22/28	22/28	
		Median and IQR		
Red blood cells (10^3^/mm^3^)	4550 (4202.5–5087.5)	4585 (4167.5–5030)	4533 (4187.5–5020)	ns
Hemoglobin (g%)	12.4 (10.6–13.7)	12.3 (11.2–13.6)	12.65 (11.3–13.9)	ns
HCT %	38.6 (33.7–41)	37.6 (33.9–41.7)	38.2 (34.7–41.7)	ns
MCV fl	83 (76–89.8)	83.05 (77.9–90.5)	83.2 (76.9–87.9)	ns
MCH pg	27.36 (24.7–30.3)	27.9 (24.1–30.1)	27.7 (25.2–29.6)	ns
White blood cells (10^3^/mm^3^)	9445 (6550–11400)	7505 (5900–9591)	7265 (6050–9755)	<0.001
Platelets (10^3^/mm^3^)	299.5 (231.5–438)	274.3 (231.1–380.5)	269.5 (222.8–381.2)	ns
C-reactive protein (mg/dL)	4.57 (1.42–15)	3.05 (0.42–8.65)	3.35 (1–10.2)	<0.001
Erythrocyte sedimentation rate (mm)	20 (11.3–42.3)	15 (8–40)	16 (7.75–40)	ns
Total protein (g/dL)	6.97 (6.7–7.3)	7.1 (6.8–7.4)	7.1 (6.7–7.5)	ns
Alpha1 globulins %	4.85 (3.3–5.52)	4.3 (3–5.3)	3.87 (2.8–5)	<0.001
Alpha2 globulins %	12.2 (10.2–13.3)	11.2 (9.1–12.5)	11.1 (9.6–12.2)	<0.001
		Median (min–max)		
Disease activity score(s)				
SES-CD	18 (7–30)			
Rutgeerts	4 (2–4)			
Mayo	2 (1–3)			
CD (HBI)	3 (1–7)	0 (0–8)	0 (0–7)	=0.002
UC (pMS)	4 (1–9)	2 (0–7)	0 (0–8)	<0.001

Data are expressed as the median and interquartile range (IQR) or (min–max); *p*: significance level as tested by Friedman’s test. ns: (statistical non-significant).

**Table 2 jpm-13-00947-t002:** Changes in the serum inflammatory parameters over time.

	Friedman’s Test	Wilcoxon’s Test		
	*p*-Value	*p*-Value		
	T0–T1–T2	T0–T1	T1–T2	T0–T2
White blood cells	***p* < 0.001**	**0.003**	0.754	**0.002**
C-reactive protein	***p* < 0.001**	**0.047**	0.539	**0.046**
Alpha1 globulins	***p* < 0.01**	0.217	0.201	**0.008**
Alpha2 globulins	***p* < 0.001**	**0.001**	0.566	**0.002**

Changes in serum inflammatory parameters over time (T0–T1–T2) and between paired observation times (T1 vs. T0; T2 vs. T1; T2 vs. T0) as tested by Friedman’s test and Wilcoxon’s test, respectively. Significant (*p* < 0.05) variations are reported in bold.

**Table 3 jpm-13-00947-t003:** Changes between the three time points (T1 vs. T0; T2 vs. T1; T2 vs. T0) of the perceived HRQoL as measured by the SF-36, FACIT-GH, and WPAI:GH questionnaires, as tested by Wilcoxon’s test. Significant (*p* < 0.05) variations are reported in bold. SF-36 = Short Form 36; FACIT-GH = Global Functional A.

	T0-T1		T1-T2		T0-T2	
	Wilcoxon’s Test—SF-36					
	Δ%	*p*-value	Δ%	*p*-value	Δ%	*p*-value
1. Physical role and functioning	27.2	*p* < 0.001	3.21	*p* = 0.182	30.4	*p* < 0.001
2. Role and physical health	100.9	*p* < 0.001	21.2	*p* = 0.036	122.1	*p* < 0.001
3. Mental health	69.5	*p* < 0.005	14.8	*p* = 0.377	84.4	*p* < 0.001
4. Vitality	35.3	*p* < 0.001	3.56	*p* = 0.091	38.9	*p* < 0.001
5. Emotional role	29.5	*p* < 0.001	6.81	*p* = 0.006	36.3	*p* < 0.001
6. Social functioning	33.9	*p* < 0.001	6.59	*p* = 0.006	40.5	*p* < 0.001
7. Body pain	51.3	*p* < 0.001	−2.06	*p* = 0.577	49.3	*p* < 0.001
8. General health	51.8	*p* < 0.001	−0.61	*p* = 0.703	51.2	*p* < 0.001
	Wilcoxon’s Test—FACIT:GH					
	Δ%	*p*-value	Δ%	*p*-value	Δ%	*p*-value
1. Physical well-being	25.7	*p* < 0.001	4.25	*p* < 0.001	31.1	*p* = 0.000
2. Social/family well-being	−2.31	*p* = 0.411	2.37	*p* = 0.916	0	*p* = 0.419
3. Emotional well-being	22.6	*p* < 0.001	1.64	*p* = 0.233	24.56	*p* < 0.001
4. Functional well-being	18.2	*p* < 0.001	4.78	*p* = 0.216	23.84	*p* < 0.001
5. Additional concerns (FACIT-FS)	30.9	*p* < 0.001	1.15	*p* = 0.180	32.4	*p* < 0.001
6. FACIT-G	15.3	*p* < 0.001	3.93	*p* < 0.001	19.8	*p* < 0.001
7. FACIT-GH (FACIT-G + FACIT-FS)	20.4	*p* < 0.001	2.94	*p* = 0.015	23.9	*p* < 0.001
	Wilcoxon’s Test—WPAI:GH					
	Δ%	*p*-value	Δ%	*p*-value	Δ%	*p*-value
2. Work time missed (health)	−50.9	*p* < 0.05	−15.4	*p* = 0.285	−58.4	*p* < 0.05
3. Work time missed (other)	−21.6	*p* = 0.655	5.47	*p* = 0.317	−17.3	*p* = 0.285
4. Effective worked hours	−11.6	*p* < 0.007	5.94	*p* = 0.027	−6.36	*p* < 0.001
5. Impairment at work	−40.1	*p* < 0.008	−11.3	*p* = 0.043	−46.9	*p* < 0.001
6. Regular activity impairment	−37.7	*p* < 0.000	−8.8	*p* = 0.088	−43.2	*p* < 0.001

**Table 4 jpm-13-00947-t004:** Spearman’s correlations between the serum parameters and the SF-36, FACIT-F, and WPAI:GH subscales. Significant (*p* < 0.05) correlations are reported in bold.

Spearman’s Test								
	SF-36							
	ΔWBC	ΔNeu	ΔLimph	ΔCRP	Δalpha1	ΔHb	ΔMCV	ΔMCH
1. ΔPhysical functioning				rs 0.301 *p* = 0.038				
2. ΔPhysical health	rs 0.418 *p* = 0.004	rs 0.555 *p* = 0.002	rs −0.496 *p* = 0.014	rs 0.583 *p* = 0.003				
3. ΔMental health								
4. ΔVitality								
5. ΔEmotional role				rs 0.315 *p* = 0.026		rs −0.311 *p* = 0.028	rs −0.248 *p* < 0.05	rs −0.349 *p* < 0.01
6. ΔSocial functioning								
7. ΔBody pain								
8. ΔGeneral health							rs −0.254 *p* < 0.05	rs −0.266 *p* < 0.05
	FACIT-F							
	ΔWBC	ΔNeu	ΔLimph	ΔCRP	Δalpha1	ΔHb	ΔMCV	ΔMCH
1. ΔPhysical well-being				rs 0.370 *p* = 0.008			rs −0.348 *p* < 0.01	
2. ΔSocial/family well-being								
3. ΔEmotional well-being								
4. ΔFunctional well-being								
5. ΔAdditional concerns (FACIT-FS)								
6. ΔFACIT-G				rs 0.303 *p* = 0.032			rs −0.344 *p* = 0.014	
7. ΔFACIT-GH							rs −0.369 *p* = 0.008	
	WPAI:GH							
	ΔWBC	ΔNeu	ΔLimph	ΔCRP	Δalpha1	ΔHb	ΔMCV	ΔMCH
2. ΔWork time missed (health)							rs 0.634 *p* < 0.001	rs 0.857 *p* < 0.001
3. ΔWork time missed (other)	rs −0.515 *p* = 0.002				rs 0.488 *p* = 0.003			
4. ΔEffective worked hours	rs −0.437 *p* = 0.003							
5. ΔImpairment at work	rs −0.496 *p* = 0.002							
6. ΔRegular activity impairment								

**Table 5 jpm-13-00947-t005:** Interdependence analysis among the serum parameters, which was performed by the Spearman’s test (baseline).

Serum Parameters		Spearman’s Test
Alpha1 globulins	WBC	rs −0.379, *p* = 0.007
Alpha1 globulins	ESR	rs 0.458, *p* = 0.001
Alpha1	Alpha2 globulins	rs 0.501, *p* = 0.000
ESR	CRP	rs 0.305, *p* = 0.031
MCV	MCH	rs 0.757, *p* = 0.000
MCV	Hb	rs 0.315, *p* = 0.026
MCH	Hb	rs 0.477, p= 0.000

**Table 6 jpm-13-00947-t006:** Perceived HRQoL change over time and treatment comparisons.

	Infliximab					W	Vedolizumab					W	MW
SF-36	T0	T1	T2	Δ_1_%	Δ_2_%	*p*	T0	T1	T2	Δ_1_%	Δ_2_%	*p*	*p*
4.	40 (36.2)	60 (22.5)	63.6 (25)	49.7	60.8	0.000	42.5 (18.75)	55 (22.5)	50 (15)	25.8	23.1	0.001	0.001
6.	43.75 (40.6)	68.8 (37.5)	75 (37.5)	48.0	61.0	0.001	50 (25)	50 (37.5)	52.5 (32.5)	23.0	24.7	0.004	0.001
7.	35 (53.1)	67.5 (30.6)	77.5 (45)	65.8	77.2	0.000	45 (31.8)	55 (30)	53.7 (19.4)	40	28.2	0.006	0.001
FACIT-F	T0	T1	T2	Δ_1_%	Δ_2_%	*p*	T0	T1	T2	Δ_1_%	Δ_2_%	*p*	*p*
1.	17.5 (13.5)	24 (3)	25 (5)	37.4	45.6	0.000	20 (9.5)	23.5 (6)	24 (5.5)	17.6	21	0.000	0.001
5.	31.5 (21.5)	42 (10.5)	43 (10.2)	43	50.7	0.000	30.5 (16.2)	38.5 (8.5)	39 (13.2)	22.2	19.2	0.001	0.001

Main HRQoL SF-36 and FACIT items at baseline (T0) and after 14 weeks (T2); delta (%) change over time; W: Wilcoxon test (T2-T0); MW: Mann–Whitney test (infliximab vs. vedolizumab). Δ_1_%: delta percentage between T0 and T1; Δ_2_%: delta percentage between T0 and T2. SF-36 items: 4: vitality; 6: social functioning; 7: body pain. FACIT items: 1: physical well-being; 5: additional concerns (FACIT-FS).

## Data Availability

The raw data and analysis are available upon any reasonable request.

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
