# Peer review of "Quality of Life (QoL) in Patients with Chronic Inflammatory Bowel Diseases: How Much Better with Biological Drugs?"

_jpm, 2023, doi:10.3390/jpm13060947_

Round 1

Reviewer 1 Report

This paper describes the change in HRQoL and markers of inflammation in IBD patients undergoing biological drugs (infliximab or vedolizumab). Although, this study is well performed and nicely described, it lacks innovative results. As many more, even pivotal trials, has shown this in lager study populations and in comparison, to placebo. This is a major limitation.

Major comments

1. Line 232. Please indicate if all patients started biological therapy due to active luminal disease confirmed on endoscopy as the HBI and pMS at baseline indicates that many patients (especially CD patients) are in clinical remission at time of start of biological therapy. As a Harvey-Bradshaw Index Score < 5 and Partial Mayo Index Score < 2 is considered to be in remission.

In addition add median HBI and pMS in table 1.

2. Please explain the high number of non-workers in this population (36%), as this is not representative for the whole IBD population.

3. In general the discussion should focus more on what is new in this study in comparison to other studies as many pivotal trails have used HRQL during induction therapy in larger populations and in comparison to placebo. For instance also summarized in following reference that is missing in the list.

Health-Related Quality of Life of Patients Treated with Biological Agents and New Small-Molecule Drugs for Moderate to Severe Crohn's Disease: A Systematic Review.

Aladraj H, Abdulla M, Guraya SY, Guraya SS.J Clin Med. 2022 Jun 28;11(13):3743. doi: 10.3390/jcm11133743.

Minor comments

1. Line 147: Should ESV be ESR? “Hematological and clinical biochemistry parameters (ESV, CRP, alfa1-alfa2 globulin 147 and blood count) were measured after a 12-hours fast during the same timepoints.”

2. Line 189: Wilcoxon test should be replaced by Wilcoxon signed-rank sum test to be clearer.

3. line 207: Please adapted following sentence. ‘A large amount of the participants (76%, n=38) didn’t never undergo IBD 207 surgical treatment, and no patient needed for surgery during the observation’. This should be: “A large amount of the participants (76%, n=38) never underwent IBD surgery, nor in the 14 weeks after start of biological treatment.”

4. Please be consisted in the use of pMS instead of PMS.

5. Please correct for multiple testing when looking for associations between serum biomarkers and scores of the individual subscales of SF-36 and FACIT-F questionnaires.

6. Please replace biohumoral to serum markers or inflammatory biomarkers

7. Maybe add the results of Δ of all SF-36 or FACIT to each other as a supplementary table.

8. In abstract T2 in noted as week 14, however in the manuscript itself it is noted as week 12. Please correct, as normally week 14 will be start of maintenance therapy and not week 12.

9. Please add p-value of the changes in scores in IBD patients treated with infliximab versus vedolizumab in each subfigure of figure 4. As you stated that patients receiving infliximab experienced a more pronounced improvement of perceived HRQoL.

10. Please also correlate SF-36 and FACIT-F subscales with pMS and HBI and add this results in the manuscript. As you state: “improvement in HRQoL seems to be closely related to treatment response and a positive correlation between clinical response and improvement in HRQoL was seen.” And adjusted paragraph 3 of the discussion accordingly.

11. Were the patients group treated with infliximab and vedolizumab statistically different at base line based on clinical or biological disease activity index? As this could also explain difference in Δ SF-36 and FACIT-F subscales.

12. Please remove following sentence (line 323-328): In line with current knowledge, we found high levels of inflammatory serum indexes at baseline; 12 weeks after the start of the biological treatment administration, a significant reduction of CRP, WBC, alpha1 and alpha2 globulins serum levels emerged, as previously argued. The same results were precociously observed at the first timepoint (6 weeks) for all the inflammatory biomarkers, without reaching statistical significance only for alpha1 globulins parameter. This is not the main message of this study and previously confirmed in lager studies.

13. Please remove following sentence: “Our study demonstrates that early start of biologic therapy can result in a better clinical response, expressed in terms of reduced HBI and pMS scores, as just argued; concomitantly, in our setting of IBD patients, we observed a significant reduction in steroid use. Pivotal trials already show the benefit of start biological treatment and its effect on steroids and you certainly didn’t show the benefit or early start of treatment.

14. Please add age at diagnosis and age at time of start of biological therapy in table 1. As well as the number of biologicals previously started.  

15. Please move following sentence from paragraph 3 in the discussion to line 386 (when discussing difference between infliximab and vedolizumab). “Although the small sample size does not allow to generalize the results, we found a difference in the impact of the treatment type on some HRQoL items, with a greater improvement related to infliximab administration with respect to vedolizumab, at least within the first 12 weeks.” The rest of paragraph 3 is currently not confirmed with your data and should be removed.

16. You state at line 402 that clinical disease severity at baseline is lacking. However, at T0 you have HBI and pMS values available as indication for clinical disease severity. How do you explain this?

17. Please add reference for following statements: Few reports showed the possible link and correlation between systemic inflammation (evaluated through serum biomarkers) and specific disease (eg. cancer, post traumatic stress syndrome, SARS-CoV-2, …). “Some of these papers have been explicitly focused on the relationship between CRP and HRQoL. Other studies also focused on the link between HRQoL and interleukins”.

18. Please remove following sentences in the conclusion section as these are not validated: “Moreover, we confirmed the possible benefit of reducing steroids not only in terms of side effects but also of HRQoL. Our findings suggest clinicians to perform an earlier assessment and monitoring of HRQoL, with the dual purpose of evaluate their clinical response and remission”.

Author Response

This paper describes the change in HRQoL and markers of inflammation in IBD patients undergoing biological drugs (infliximab or vedolizumab). Although, this study is well performed and nicely described, it lacks innovative results. As many more, even pivotal trials, has shown this in lager study populations and in comparison, to placebo. This is a major limitation.

Major comments

  1. Line 232. Please indicate if all patients started biological therapy due to active luminal disease confirmed on endoscopy as the HBI and pMS at baseline indicates that many patients (especially CD patients) are in clinical remission at time of start of biological therapy. As a Harvey-Bradshaw Index Score < 5 and Partial Mayo Index Score < 2 is considered to be in remission.

R: Thank you for this advice, we agree with this comment. In the revised manuscript we added the detailed description of endoscopic activity at baseline of patients enrolled (you can now see it added in results section, lines 213-218). Please, see also endoscopic scores added in methods (lines 142-145).

Recent endoscopy (latest six months) at baseline was available in 42/50 patients (84%) and active disease (moderate-to-severe) was confirmed in all of them. In details, patients with CD had a median SES-CD of 18 (7-30) and a median Rutgeerts score of 4 (2-4), and patients with UC had a median endoscopic Mayo score of 2 (2-3). Recent Endoscopy was not available for 7 patients. Among them, two patients had instrumental signs (MRI or bowel ultrasound) of increased disease activity.

1.1 In addition, add median HBI and pMS in table 1.

R: We revised the table accordingly.

  1. Please explain the high number of non-workers in this population (36%), as this is not representative for the whole IBD population.

R: We agree with the Reviewer, however our data are in line with the employment data from the Italian National Institute of Statistics (ISTAT) of the Sicilian population relative to the year of participants’ recruitment. In fact, the number of Sicilian households with no one employed was markedly higher than the national average in 2018 (in details, 32.5% versus 18.4% percent in the whole Italian area, n. 479/1474). We added a clarification in the revised manuscript to avoid any data misinterpretation.

  1. In general the discussion should focus more on what is new in this study in comparison to other studies as many pivotal trails have used HRQL during induction therapy in larger populations and in comparison to placebo. For instance also summarized in following reference that is missing in the list.

Health-Related Quality of Life of Patients Treated with Biological Agents and New Small-Molecule Drugs for Moderate to Severe Crohn's Disease: A Systematic Review.

Aladraj H, Abdulla M, Guraya SY, Guraya SS.J Clin Med. 2022 Jun 28;11(13):3743. doi: 10.3390/jcm11133743.

R: We thank the Reviewer for this precious suggestion. We tried to better underline the strengths of our research in the Discussion. Also, we have now updated the literature with suggested research and deeply revised the discussion session accordingly.

Minor comments

  1. Line 147: Should ESV be ESR? “Hematological and clinical biochemistry parameters (ESV, CRP, alfa1-alfa2 globulin 147 and blood count) were measured after a 12-hours fast during the same timepoints.”

R: the Reviewer is right; in the original manuscript ESR (erythrocytes sedimentation rate) was incorrectly reported as ESV; this mistake is now fixed.

  1. Line 189: Wilcoxon test should be replaced by Wilcoxon signed-rank sum test to be clearer.

R: We thank the Reviewer for this suggestion; the text was revised accordingly.

  1. line 207: Please adapted following sentence. ‘A large amount of the participants (76%, n=38) didn’t never undergo IBD surgical treatment, and no patient needed for surgery during the observation’. This should be: “A large amount of the participants (76%, n=38) never underwent IBD surgery, nor in the 14 weeks after start of biological treatment.”

R: The sentence has been adapted as suggested.

  1. Please be consisted in the use of pMS instead of PMS.

R: The abbreviation of partial Mayo score was rewritten in the revised version of the manuscript as suggested. Moreover, wellbeing is now changed to well-being throughout the text.

  1. Please correct for multiple testing when looking for associations between serum biomarkers and scores of the individual subscales of SF-36 and FACIT-F questionnaires.

R: In the revised manuscript we preferred to show the interrelationships (as evaluated by Spearman’s test) between the delta (T2-T0) of the considered serum markers with the delta of QoL questionnaires items; this way, no correction for multiple comparisons is needed.

  1. Please replace biohumoral to serum markers or inflammatory biomarkers.

R: We revised the text according to this Reviewer’ suggestion.

  1. Maybe add the results of Δ of all SF-36 or FACIT to each other as a supplementary table.

R: We thank the Reviewer for the advice; since a number of items were already summarized in table 3, we completed it with the missing ones; as regards Δ correlations with each other, we prepared two new tables as S1 and S2.

  1. In abstract T2 in noted as week 14, however in the manuscript itself it is noted as week 12. Please correct, as normally week 14 will be start of maintenance therapy and not week 12.

R: We thank the Reviewer for this precious suggestion. We corrected the text accordingly.

  1. Please add p-value of the changes in scores in IBD patients treated with infliximab versus vedolizumab in each subfigure of figure 4. As you stated that patients receiving infliximab experienced a more pronounced improvement of perceived HRQoL.

R: Figure 4 was revised adding p-value of the changes in scores in IBD patients treated with infliximab versus vedolizumab in each paneled chart, as suggested by Reviewer.

  1. Please also correlate SF-36 and FACIT-F subscales with pMS and HBI and add this results in the manuscript. As you state: “improvement in HRQoL seems to be closely related to treatment response and a positive correlation between clinical response and improvement in HRQoL was seen.” And adjusted paragraph 3 of the discussion accordingly.

R: We thank the Reviewer for this precious suggestion. Correlations between SF-36 and FACIT-F subscales and pMS and HBI are now added in the revised manuscript. We extensively revised the text accordingly.

  1. Were the patients group treated with infliximab and vedolizumab statistically different at base line based on clinical or biological disease activity index? As this could also explain difference in Δ SF-36 and FACIT-F subscales.

R: Thank you for your valuable suggestion; we repeated the test in order to include any potential statistical difference in HRQoL scores at baseline, verifying each item separately; as you can see in the report table embedded below, we found no significant difference at baseline in the considered items.

sf36_1

sf36_2

sf36_3

sf36_4

sf36_5

sf36_6

sf36_7

sf36_8

facit1

facit2

facit3

facit4

facit5

facit6

facit7

wpai_1

wpai_2

wpai_3

wpai_4

wpai_5

wpai_6

MW-U

245,5

295,5

297,5

295,5

299,0

276,5

297,0

291,5

310,5

295,5

284,0

297,0

276,5

303,5

291,5

286,0

54,0

43,0

62,0

58,0

271,5

W

545,5

595,5

648,5

595,5

650,0

627,5

597,0

591,5

610,5

595,5

635,0

597,0

627,5

654,5

642,5

586,0

145,0

98,0

117,0

113,0

622,5

Z

-1,296

-,348

-,283

-,322

-,253

-,705

-,293

-,400

-,029

-,321

-,546

-,292

-,690

-,165

-,398

-,585

-,777

-1,400

-,187

-,439

-,790

p

,195

,728

,777

,747

,800

,481

,769

,689

,977

,748

,585

,770

,490

,869

,690

,559

,437

,161

,852

,661

,429

Mann Whitney comparison between CD e UC patients at baseline

  1. Please remove following sentence (line 323-328): In line with current knowledge, we found high levels of inflammatory serum indexes at baseline; 12 weeks after the start of the biological treatment administration, a significant reduction of CRP, WBC, alpha1 and alpha2 globulins serum levels emerged, as previously argued. The same results were precociously observed at the first timepoint (6 weeks) for all the inflammatory biomarkers, without reaching statistical significance only for alpha1 globulins parameter. This is not the main message of this study and previously confirmed in lager studies.

R: The above sentence was removed in the revised manuscript as requested.

  1. Please remove following sentence: “Our study demonstrates that early start of biologic therapy can result in a better clinical response, expressed in terms of reduced HBI and pMS scores, as just argued; concomitantly, in our setting of IBD patients, we observed a significant reduction in steroid use. Pivotal trials already show the benefit of start biological treatment and its effect on steroids and you certainly didn’t show the benefit or early start of treatment.

R: We removed the sentence in the revised version of the manuscript.

  1. Please add age at diagnosis and age at time of start of biological therapy in table 1. As well as the number of biologicals previously started.

R: We thank the Reviewer for this advice. As requested, we have added in the revised version of the manuscript (Table 1) the median value of age at the diagnosis and age at time of start of biological therapy. No other biological drugs had been started previously by enrolled patients.

  1. Please move following sentence from paragraph 3 in the discussion to line 386 (when discussing difference between infliximab and vedolizumab). “Although the small sample size does not allow to generalize the results, we found a difference in the impact of the treatment type on some HRQoL items, with a greater improvement related to infliximab administration with respect to vedolizumab, at least within the first 12 weeks.” The rest of paragraph 3 is currently not confirmed with your data and should be removed.

R: The text was revised according to the Reviewer’s comments.

  1. You state at line 402 that clinical disease severity at baseline is lacking. However, at T0 you have HBI and pMS values available as indication for clinical disease severity. How do you explain this?

R: We thank the Reviewer for this precious suggestion; also according to another your suggestion, in the revised manuscript also endoscopic activity is now added; we changed the text in line with these considerations.

  1. Please add reference for following statements: Few reports showed the possible link and correlation between systemic inflammation (evaluated through serum biomarkers) and specific disease (eg. cancer, post traumatic stress syndrome, SARS-CoV-2, …). “Some of these papers have been explicitly focused on the relationship between CRP and HRQoL. Other studies also focused on the link between HRQoL and interleukins”.

R: The reference list has been updated according to the Reviewer ‘suggestion.

  1. Please remove following sentences in the conclusion section as these are not validated: “Moreover, we confirmed the possible benefit of reducing steroids not only in terms of side effects but also of HRQoL. Our findings suggest clinicians to perform an earlier assessment and monitoring of HRQoL, with the dual purpose of evaluate their clinical response and remission”.

R: The above-mentioned sentences were removed in the revised manuscript accordingly.

Reviewer 2 Report

Currently, IBD treatment endgoals are based on various score systems, such as Crohn's disease activity index (CDAI), Inflammatory Bowel Disease Questionnaire (IBDQ) and Harvey-Bradshaw Index (HBI) but also sometimes together with scores based on colonoscopy data such as Simple Endoscopic Score for Crohn Disease (SES-CD) and the Mayo Score. The previously mentioned scores are robust and adequate for routine use, but are limited in their consideration for the patient's overall health and wellbeing (except IBDQ). The authors attempted to validate a SF-36 questionairre variant, FACIT questionairre and WPAI:GH on a cohort with fifty IBD (CD and UC) patients treated with infliximab or vedolizumab. As such, the endpoint may be shifted from highly clinical scoring systems to ones with patient wellbeing in mind.

The study design is sound, the results suggest that the new questionnaires are adequate to monitor patient wellbeing during infliximab and vedolizumab treatment. The statistical power seems sufficient as well. In addition, the findings show that there is correlation with HBI and other criteria. Moreover, variable independance was tested as well.

I suggest the article to be published with minor revisions:

(1) Correct Figure 1. It appears to be cut off in the presentted manuscript PDF.

(2) Any p values written as "p 0,000" to be substituted with "p < 0.001". In addition, pay attention to comma use for decimal spaces: English generally uses dots as decimal spaces.

(3) If I am not mistaken, the lower bounds of error bars on box plots are not visible.

Author Response

Reviewer#2

Currently, IBD treatment endgoals are based on various score systems, such as Crohn's disease activity index (CDAI), Inflammatory Bowel Disease Questionnaire (IBDQ) and Harvey-Bradshaw Index (HBI) but also sometimes together with scores based on colonoscopy data such as Simple Endoscopic Score for Crohn Disease (SES-CD) and the Mayo Score. The previously mentioned scores are robust and adequate for routine use, but are limited in their consideration for the patient's overall health and wellbeing (except IBDQ). The authors attempted to validate a SF-36 questionairre variant, FACIT questionairre and WPAI:GH on a cohort with fifty IBD (CD and UC) patients treated with infliximab or vedolizumab. As such, the endpoint may be shifted from highly clinical scoring systems to ones with patient wellbeing in mind.

The study design is sound, the results suggest that the new questionnaires are adequate to monitor patient wellbeing during infliximab and vedolizumab treatment. The statistical power seems sufficient as well. In addition, the findings show that there is correlation with HBI and other criteria. Moreover, variable independance was tested as well.

I suggest the article to be published with minor revisions:

(1) Correct Figure 1. It appears to be cut off in the presentted manuscript PDF.

R: We’re really sorry, but we can see Figure 1 right as we expect. Could the Reviewer please better detail the issue?

(2) Any p values written as "p 0,0" to be substituted with "p < 0.001". In addition, pay attention to comma use for decimal spaces: English generally uses dots as decimal spaces.

R: Thank you very much for the advice; we checked and tried to fix all the p-values throughout the text.

(3) If I am not mistaken, the lower bounds of error bars on box plots are not visible.

R: We’re sorry for this mistake; as you can see, data on figure are depicted through a histogram chart, and not a boxplot; we had erroneously described it in the former version of the manuscript.